# In Silico Study of Camptothecin-Based Pro-Drugs Binding to Human Carboxylesterase 2

**DOI:** 10.3390/biom14020153

**Published:** 2024-01-27

**Authors:** Frank Beierlein, Anselm H. C. Horn, Heinrich Sticht, Andriy Mokhir, Petra Imhof

**Affiliations:** 1Department for Chemistry and Pharmacy, Computer Chemistry Center, Friedrich-Alexander University Erlangen Nürnberg (FAU), Nägelsbachstraße 25, 91052 Erlangen, Germany; frank.beierlein@fau.de; 2Erlangen National High Performance Computing Center (NHR@FAU), Friedrich-Alexander University Erlangen Nürnberg (FAU), Martensstraße 1, 91058 Erlangen, Germany; anselm.horn@fau.de; 3Institute of Biochemistry, Friedrich-Alexander University Erlangen Nürnberg (FAU), Fahrstraße 17, 91054 Erlangen, Germany; heinrich.sticht@fau.de; 4Department for Chemistry and Pharmacy, Institute for Organic Chemistry, Friedrich-Alexander University Erlangen Nürnberg (FAU), Nikolaus-Fiebiger-Straße 10, 91058 Erlangen, Germany; andriy.mokhir@fau.de

**Keywords:** pro-drug, camptothecin, human carboxylesterase, homology model, docking, molecular dynamics simulations

## Abstract

Pro-drugs, which ideally release their active compound only at the site of action, i.e., in a cancer cell, are a promising approach towards an increased specificity and hence reduced side effects in chemotherapy. A popular form of pro-drugs is esters, which are activated upon their hydrolysis. Since carboxylesterases that catalyse such a hydrolysis reaction are also abundant in normal tissue, it is of great interest whether a putative pro-drug is a probable substrate of such an enzyme and hence bears the danger of being activated not just in the target environment, i.e., in cancer cells. In this work, we study the binding mode of carboxylesters of the drug molecule camptothecin, which is an inhibitor of topoisomerase I, of varying size to human carboxylesterase 2 (HCE2) by molecular docking and molecular dynamics simulations. A comparison to irinotecan, known to be a substrate of HCE2, shows that all three pro-drugs analysed in this work can bind to the HCE2 protein, but not in a pose that is well suited for subsequent hydrolysis. Our data suggest, moreover, that for the irinotecan substrate, a reactant-competent pose is stabilised once the initial proton transfer from the putative nucleophile Ser202 to the His431 of the catalytic triad has already occurred. Our simulation work also shows that it is important to go beyond the static models obtained from molecular docking and include the flexibility of enzyme–ligand complexes in solvents and at a finite temperature. Under such conditions, the pro-drugs studied in this work are unlikely to be hydrolysed by the HCE2 enzyme, indicating a low risk of undesired drug release in normal tissue.

## 1. Introduction

A pro-drug is a molecule that is pharmacologically inactive but can be converted to an active drug inside the body [1]. Ester pro-drugs are frequently administered orally instead of the actual drug so as to achieve better absorption of the esters, which are typically less polar, and thus more permeating, than the actual drugs. These drugs are then usually activated through cleaving the ester bond of the pro-drug by carboxylesterases [2,3,4], which are found in the liver or the small intestine, for example. The polar products of this reaction, carboxylic acids and alcohols, one of which then acts as the actual drug, are then distributed in the bloodstream and are finally subject to excretion. Esterases are also sometimes involved in the detoxification of compounds [2].

In our previous work [5], we presented an aminoferrocen-based pro-drug that can be used to release camptothecin (CPT) and other cytostatics via hydrolysis of an ester bond under cancer-specific conditions, i.e., increased levels of reactive oxygen species (ROS) and basic pH in the mitochondria lumen/endoplasmatic reticulum. In this way, differences in the chemical or biological properties of different tissues or bio-compartments [5,6] are exploited such that pro-drugs are specifically activated in their desired target tissue. This strategy, however, can be invalidated if enzymes, which are present in several different tissues and compartments, accelerate the activation reaction.

In mammals, several sub-types of carboxylesterases are known, e.g., human carboxylesterases 1 (HCE1) and 2 (HCE2) and rabbit carboxylesterase (RCE). In humans, HCE1 is predominantly found in the liver, while HCE2 is found in the small intestine in higher concentrations [2,4,7]. In a cell, carboxylesterases are mainly located in the luminal side of the membrane of the endoplasmatic reticulum (ER) [2,4,8]. It should be noted that expression of carboxylesterases is affected by many factors; thus, a large variability among individuals and tissues is observed [2,9]. Although carboxylesterases are known to be relatively promiscuous enzymes, HCE1’s preference for substrates with a larger acyl moiety and a smaller alcohol rest has been observed, while for HCE2, substrates with larger alcohol moieties and smaller acyl rests are preferred [2,4,8,10].

While several X-ray structures have been reported for HCE1 and RCE, e.g., [11,12], at the time of writing, no structures of HCE2 are available in the PDB database, and so far only comparative/homology model structures are available for HCE2 [8,13]. The biological unit of HCE1 has been described as a trimer or a hexamer formed by glycosylated monomers [11], with the active sites at a sufficient distance from the complex interface. For HCE2, a monomeric structure of the biologically active form is assumed [8,10]. A protein loop that can act as a putative lid on the binding pocket in HCE1 is missing in HCE2 [8]. The entry channel of the substrate-binding pocket is flanked by two helices at the protein surface, which exhibit some conformational flexibility and may possibly further regulate access to the binding pocket [3]. At the bottom of the deep binding pocket, a catalytic triad formed by serine, histidine, and glutamate is found (see Section 3.1). These residues divide the binding pocket into two sub-pockets, one for the acyl moiety of the substrate and one for the alcohol part. In HCE1, the acyl sub-pocket is lined by more nonpolar residues and the alcohol sub-pocket by more polar ones [3,8]. For HCE2, an additional part of the acyl sub-pocket is observed, which is lined by several glutamate residues that can attract positively charged parts of a ligand [8]. For both enzymes, a “side door” has been described that may be involved in the release of smaller product molecules [4,8,12].

A putative reaction scheme for carboxyl ester bond cleavage by HCE2 according to references [3,4] is shown in Figure 1. After attack of the Ser202–OG atom on the ester carbonyl atom of the substrate, a tetrahedral intermediate is formed, which is stabilised by interactions of the negatively charged carbonyl oxygen atom with the oxyanion hole formed by the backbone amide hydrogen atoms of Gly123, Ala124, and Ala203. After elimination of the alcohol, the complex between Ser202 and the acyl part of the ligand is hydrolysed in a second step by a water molecule, resulting in the restoration of the catalytic triad to its original form and the dissociation of the ligand acyl moiety as carboxylic acid.

It should be noted that esterases can also perform transesterification [14] if the attacking water molecule is replaced by an alcohol, and that a protonated acyl rest of the ligand can interact with the Glu-rich sub-pocket in HCE2 and thus possibly act as an inhibitor [6,8,15].

Overviews of known substrates of HCE1 and HCE2 are given in the literature [2,4]. Also, natural substances and xenobiotiocs (e.g., insecticides) can bind as substrates or inhibitors, and are detoxified or affect drug metabolism [8,16,17,18,19]. Among several other routinely administered drugs, irinotecan ({7-ethyl-10-[4-(1-piperidino)-1-piperidino]} carbonyloxy-camptothecin) is a well-known substrate of HCE2 and RCE [2,6,12,20]. It is mainly used to treat colon cancer and is a pro-drug that releases SN-38 (7-ethyl-10-hydroxycamptothecin) as its active metabolite, an analogue of camptothecin, which is an inhibitor of topoisomerase I and thus causes apoptosis by inhibiting DNA transcription and replication.

Here, we investigate whether human carboxylesterase 2, which hydrolyses esters with relatively large alcohol and small acyl moieties, may be involved in the activation of camptothecin ester pro-drugs (amino-triazol-alkyl-acid camptothecin esters). To this end, we have built structural models of HCE2 in apo form and complexed to several variants of the camptothecin ester pro-drug, L2, L5, and L8 (see Figure 2), and to the known substrate irinotecan (It) [2,4,21] as a control.

This paper is structured as follows: we first describe the modelling of the pro-drug ligands and irinotecan in water (Section 2.1), the construction of the homology model of HCE2 (Section 2.2), followed by the docking protocol (Section 2.3) and the details of the molecular dynamics simulations (Section 2.4) and their analysis (Section 2.5). We then present our homology model of HCE2 in comparison to the structure of the HCE1 protein from the simulations in solution and with two different protonation states of active site residues (Section 3.1). Section 3.2 presents a comparison of the protein–ligand complexes, analysing the conformations of the catalytic triad; Section 3.3 describes the ligand poses; and Section 3.4 discusses important interactions between protein and ligands. A discussion of our results in light of putative pro-drug ester hydrolysis by HCE2 is presented in Section 4. As a conclusion, we have successfully modelled HCE2 with bound ligands, allowing us to study the intricacies of the architecture of the catalytic triad and to qualitatively evaluate the suitability of the pro-drugs as a substrate to HCE2. Our data suggest that, while binding of the pro-drug ligands to HCE2 is likely, their pose is unfavourable for ester hydrolysis via the HCE2 enzyme.

## 2. Materials and Methods

### 2.1. Ligands in Water

Molecular dynamics simulations of the ligand compounds L2, L5, and L8 and irinotecan were performed with Amber 20 [22] pmemd.cuda, using Gaff1.81 [23,24] with RESP [25,26] charges based on calculations with Gaussian 16 [27] (HF/6-31G*//B3LYP/6-31G* [28,29,30,31,32,33,34,35,36,37,38], geometry optimisations were performed in water, using a polarisable continuum model) [39,40], in agreement with the Amber force fields [26,41]. The resulting parameter files are available in the Appendix A. The solutes were solvated with TIP3P [42] water (truncated octahedral boxes exceeding the molecule dimensions by 20 Å in either direction); in the case of the protonated compounds, one Cl^−^ ion was added to neutralise the system [43]. After initial geometry optimisation (first 5000 steps with restraints (50 kcal mol^−1^ Å^−2^) on the ligands, then 5000 optimisation steps without restraints, switching from steepest descent to conjugate gradients after 500 steps in either case), the solvated systems were heated to 310 K during a 500 ps simulation with weak restraints (10 kcal mol^−1^ Å^−2^) on the ligands in the NVT ensemble. After that, 1000 ns unrestrained NPT Langevin dynamics with a time step of 2 fs were performed for each simulation system at 310 K and 1 bar (weak pressure coupling, isotropic position scaling, pressure relaxation time 2 ps, collision frequency 2 ps^−1^). SHAKE constraints were applied to bonds involving hydrogen [44]. Periodic boundary conditions were used throughout and the distance cutoff for all non-bonding interactions was set to 10 Å. Long-range electrostatics were described by the particle mesh Ewald method [45,46]. For van der Waals interactions beyond those included in the direct sum, a continuum model correction for energy and pressure was used, as implemented in Amber. Coordinates were saved every 10 ps.

### 2.2. Protein Model

As no 3D structure was available for human carboxylesterase 2 (HCE2) in the protein data bank (PDB) [47,48] we used the Swiss-Model [49,50,51,52,53] server to prepare a suitable homology model of the enzyme. The sequence of HCE2 was obtained from the Uniprot database [54] (O00748, EST2_HUMAN) and the N-terminal signal peptide (residues 1–26) was removed. The remaining protein residues were used for a template search on the Swiss-Model server, employing both the Blast [55] and the HHBlits [56] search algorithms. The PDB [47,48] structure 5A7H [11] was chosen as a template for model generation (48.34% sequence identity based on HHBlits alignment). The model obtained from Swiss-Model was thoroughly checked using WhatIf [57]; flips of glutamines, asparagines and histidines, histidine protonation states, and disulfide bridges between cysteines were investigated. In the active site, histidine 431 of the catalytic triad (Ser202, His431, and Glu319) was treated with protonation at the ND1 atom, i.e., as HID. For initial geometry optimisation with Amber 16 [58], the protein (treated by the ff14SB force field [59]) was solvated in a truncated octahedron of SPC/E water [60], exceeding the protein by 15 Å in either direction, and neutralised with sodium counter ions [43]. The solvated protein was then geometry optimised with Amber 16 following the protocol described above (ligand simulations). The resulting structure is provided as pdb file in the Appendix A. Human carboxylesterase 1 (HCE1) in its apo form was modelled analogously to HCE2, based on PDB structure 5A7H [11].

### 2.3. Modelling
of Protein–Ligand Complexes by Docking

The optimised protein structure was then used for flexible ligand/rigid receptor docking with AutoDock 4.2.6 [61,62,63] (after removal of the water molecules and the counter ions). For the docking, interaction grids of 60 Å^3^ (for comparison also 80 Å^3^ and 100 Å^3^, grid point distance 0.375 Å, were used) were centred on the Ser202 OG atom. Gasteiger–Marsili [64] charges were used for the ligand and protein atoms, and non-polar hydrogen atoms were merged with their neighbouring heavy atoms (united atom approach with polar hydrogen atoms only). For docking, all ligand/protein preparation and analysis steps were performed by AutoDock Tools 1.5.7. [63]. A total of 25 docking runs (in problematic cases, 50) were performed using the Lamarckian genetic algorithm (LGA) [62] following the long protocol with 25 × 10^6^ evaluations; for cluster analysis, an RMSD value of 2.0 Å was chosen. From the docked poses, initial structures for subsequent MDs with Amber were chosen using the following criteria: (1) a low estimated free energy of binding, (2) a high cluster population, and (3) a short distance between Ser202–OG and the ester carbonyl carbon atom of the ligand. The docking poses of the most populated clusters are shown in Appendix A. The structures of the docked complexes are available in pdb format in the Appendix A.

### 2.4. Molecular Dynamics
Simulations of the Protein–Ligand Complexes

The protein–ligand complexes obtained from docking were then used as starting structures for MD simulations with Amber20 and Amber22 [22,65] following a protocol similar to the ones explained above. After geometry optimisation, Langevin dynamics simulations of each complex were performed at 310 K for 1000 ns with snapshots saved every 100 ps. Here, TIP3P [42] water was used and, in addition to neutralising the system with sodium ions, additional NaCl was added such that a sodium concentration of 150 mM was obtained [43]. In addition to the simulations with a standard protonation state of the catalytic triad (Ser202 neutral, His431 neutral (HID), and Glu319 anionic), we also performed simulations with a deprotonated serine (HG deleted, residue name SEM) and a protonated histidine (HIP). For this purpose, we parameterised the new amino acid residue SEM; RESP charges for SEM were calculated in order to ensure parameter consistency within the set of amino acids using a standard Amber approach [25,26] with two conformations of the capped amino acid (ACE-SEM-NME) via the R.E.D. server [66,67]. All other parameters of the deprotonated amino acid SEM were kept at their ff14SB values [59]. The parameter files for SEM are available in the Appendix A. The corresponding models are labelled (SEM–HIP). The simulations of these models were performed with an initial NMR distance restraint [68] imposed on the distance between the Ser202 OG atom and the ester carbon atom of the ligand (2000 kcal · mol^−1^ Å^−2^, allowing ±0.1 Å movement from 3.75 Å), which was released after 500.5 ns. Unrestrained simulations were continued for 1000 ns.

For all models, (SER–HID) and (SEM–HIP), we performed five independent runs. Out of these, the first 100 ns of each (unrestrained) simulation was regarded as the equilibration phase and only the last 900 ns was used for data collection and analysis.

### 2.5. Analysis of Molecular Dynamics Simulation Data

Cpptraj V4.25.6 [69] from the AmberTools suite (V20.15) was used for the analyses, with further data analysis performed by custom-made Jupyter Notebooks using Numpy 1.12.6 [70] with Python 3.7.12 [71], while Matplotlib 3.5.3 [72] was used for plotting. All error estimates are the standard deviation from the mean of five individual runs per simulation setup. Hydrogen bonds were defined based on geometric criteria, i.e., a donor–acceptor distance not larger than 3.2 Å and a donor–hydrogen–acceptor angle deviating from linearity by no more than 42°. Residue interaction energies were calculated using the LIE command of Cpptraj [69] with default settings. Medoid structures (structures whose Cα positions deviate the least from the average structure of the respective run) for visualisation were obtained with Cpptraj scripts. VMD 1.9.3 [73], AutoDock Tools 1.5.7 [74], and Schrödinger Suite 2022 [75] were used for visualisation.

## 3. Results

### 3.1. Protein Model

Comparison of our homology model to the Alphafold [76,77] model of HCE2 (AF-O00748-F1-model_v3/4.pdb) shows that the two structures are very similar: The binding sites are practically identical and the root mean square deviation of the Cα atoms, calculated from a Swiss-PDB-Viewer (V4.1.0) [78] iterative magic fit using 438 Cα atoms, is 0.92 Å. As shown in Figure 3, our homology model of HCE2 is also very similar (root mean square deviation 0.42Å) to the template structure of protein HCE1.

We therefore inferred that the two apo proteins, HCE1 and HCE2, behave similarly in the course of a molecular dynamics simulation.

Figure 4 shows snapshots of the esterase apo proteins HCE1 and HCE2 in two different protonation states. The overall fold of the proteins remains stable during the MD simulations. However, in the simulation of HCE2 with neutral Ser202 and His431 (SER–HID), these two residues and Glu319 of the catalytic triad are not close enough together to allow the proton transfer that would activate the serine residue for nucleophilic attack. Once this proton transfer has happened, as mimicked by the models with deprotonated serine and protonated histidine (SEM–HIP), Glu319 and His431 stay closer together (see also Figure 5 and Appendix A). A similar observation can indeed be made for the HCE1 protein. With neutral Ser221 and His467, the distances between the catalytic residues are significantly larger than in the case of deprotonated Ser221 and protonated His467 (see Appendix A).

### 3.2. Catalytic Triad
of Protein–Ligand Complexes

A trend for the residues of the catalytic triad, similar to that found in the apo proteins, is observed in the HCE2 protein when the irinotecan ligand (It) is bound. The average distances between the residues of the catalytic triad (Figure 5), Glu319 and His431, are significantly shorter when the latter is protonated and Ser202 is deprotonated (SEM–HIP). For the systems with pro-drug ligands L2, L5, and L8 bound to the HCE2 protein, the distances between Glu319 and His431 are similar, regardless of the protonation state of Ser202 and His431. In these L2-, L5-, and L8-bound systems, however, the distances between His431 and Ser202 are shorter when Ser202 is deprotonated and His431 is protonated than when both residues are neutral. This effect is unclear in the apo and irinotecan models due to large errors which are, in particular for the irinotecan-bound model, in line with larger fluctuations in His431 and Glu319 (see Figure 6).

This imperfect catalytic triad can also be seen in the hydrogen bonds formed by its residues, as listed in Table 1. In both of the two apo proteins, no hydrogen bonds between Glu319 and neutral His431 are observed, but such a hydrogen bond is present in the (SEM–HIP) apo protein models with deprotonated Ser202 and protonated His431. In the complexes with a ligand bound to HCE2, this hydrogen bond is observed for a neutral His431 (HCE2 numbering), but is more probable when His431 is protonated. There are also observed hydrogen bonds between His431 and other glutamate residues (Glu201 and Glu434) and Asp433, the latter only in L8-bound complexes. Glu319 also forms hydrogen bonds to the nearby residue Ser228, mainly when bound to the pro-drug L8. Also, hydrogen bonds between Glu319 and Asn316 as well as between Glu319 and Gly317 (in HCE1) are observed, but with no clear dependence on the bound ligand or on the protonation state of the protein. It is interesting to note that Ser202 shows no significant probability to form hydrogen bonds to His431 with or without a bound ligand and in either protonation state. However, neutral Ser202 forms a hydrogen bond to the neighbouring Glu201 in L2-, L5-, and irinotecan-bound models, albeit with considerable errors.

### 3.3. Ligand Poses and Conformations

Figure 7 shows representative snapshots of the protein–ligand complexes in the two protonation states of the HCE2 protein, that is, with neutral Ser202 and His431 (SER–HID) and with deprotonated and protonated Ser202 and H431 (SEM–HIP), respectively. In all models, the ligand is located in the active site that is close to the catalytic triad. The ligands are also orientated in such a way that their protonated amino group points towards a group of glutamate and aspartate residues, i.e., Glu201, Glu227, Asp433, and Glu434, which allows for favourable interactions (see also Table 2). The hydrophobic ring system of the camptothecin-like moiety is located at the opposite end, pointing outwards but still positioned inside the binding pocket. For irinotecan, this means that its lactone ring is closest to the protein exterior, whereas the lactone ring of the ligands L2–L8 is more buried in the protein interior and is closer to the residues of the catalytic triad. For all bound ligands (L2, L5, and L8), the linkers are in a rather “stretched” conformation. This can also be seen in the distribution of distances between the ester C-atom and the protonated nitrogen atom of the amino group (see Appendix A), which is ∼7–12Å for the L2, L5, and L8 ligands. Irinotecan, which has no flexible linker, exhibits a distance of ∼6Å between the protonated N-atom and the ester C-atom, which is also observed for the free ligand in water. The L_*n*_ pro-drug ligands, in contrast, also exhibit folded conformations with short (<5Å) C-N distances in water, albeit with a low probability (see Appendix A), which cannot be observed when bound to the protein. As a consequence of the rather stretched conformations of the protein-bound ligands, only L2, with the shortest linker, has its ester group positioned in relative proximity to Ser202 and hence somewhat poised for attack by this residue. The positive control, irinotecan, in contrast, exhibits conformations in which its ester group is located at the “height” of Ser202 and thus poised for catalysis.

### 3.4. Protein–Ligand Interactions

As can be seen from the snapshots in Figure 7 and the average distances between the ligand and the glutamate and aspartate residues of the binding pocket (see Figure 8a,b), all ligands have at least one rather short distance from the protonated N-atom to a nearby glutamate or aspartate residue (see Figure 9). All the distances to glutamate residues are in a range that allows hydrogen bonds (∼3.5 Å). The probabilities for individual such hydrogen bonds though show rather large errors, which can be explained by fluctuations between the different acceptor residues (i.e., Glu201, Glu227, or Glu434) (see Figure 8b). Irinotecan bound to HCE2 in its (SEM–HIP) protonation state is an exception here, as it shows a highly probable, and thus stable, hydrogen bond to Glu201.

Figure 10 shows the average distance of the catalytic residue Ser202 to the C-atom of the ester group, i.e., the site of attack for the hydrolysis reaction. This distance shows large fluctuations for all protein–ligand complexes with one more populated distance of ∼6 Å and another one at ∼7.5 Å, but also distances of up to ∼10 Å in the complexes of L2, L5, and L8 ligands and for irinotecan bound to HCE2 with neutral Ser202 and His431 (SER–HID), as can be seen in the probability distributions (Figure 10). These fluctuations are in line with the variable linker lengths and therefore intramolecular C–N distances that fluctuate around ∼7–12 Å (see Appendix A), depending on the ligand. The binding pocket with the different glutamate residues that can anchor the ligand at the amino N-atom in the “upper” part of the binding pocket by hydrogen bonds thus allows some flexibility in the accommodation of the differently sized ligands. With deprotonated Ser202 and protonated His431, the distance between Ser202 and irinotecan’s ester C-atom is prominently below ∼5 Å, rendering nucleophilic attack possible.

Comparison of the electrostatic interaction energies between residues of the binding pocket and the pro-drug ligands (see Table 2) shows a clear effect of the protonation states of Ser202 and His431. While the interaction energies with the neutral forms of Ser202 and His431 (SER–HID) are small, they are considerable when these residues are charged (SEM–HIP). Deprotonated Ser202 interacts very favourably with the bound ligand, whereas the interaction with protonated His431 is unfavourable by about the same amount. The interaction energies with deprotonated SEM and protonated His431 are, within error, comparable between the different ligands. One can therefore not determine directly whether one or the other protonation state leads to more or less favourable interactions between the catalytic triad residues and the ligands. Instead, a comparison of the interaction energies with the other (glutamate and aspartate) residues of the binding pocket may be fruitful. The large fluctuations also observed in the hydrogen bonds between ligands and glutamate/aspartate residues render most interaction strengths between the ligands and the “glutamate cluster” similar. Irinotecan bound to the (SEM–HIP) model of HCE2, though, is markedly different. It has a considerably higher electrostatic interaction energy with Glu201 than all other ligands, in agreement with the strong hydrogen bond observed. This interaction, while still present in the models with neutral Ser202 and His431 (SER–HID), cannot be considered significantly different from the other ligand complexes due to the rather large errors in this quantity. On the other hand, the electrostatic interactions of irinotecan with the catalytic Glu319, but also with Glu434, are considerably weaker in the models of both protonation states of HCE2 than in the complexes of the other ligands, possibly due to the lower polarity of the tertiary amino group and the steric requirements of the ring containing the amino group in irinotecan compared to the primary amino group in the ligands L2, L5, and L8.

## 4. Discussion

The homology model of the apo protein HCE2 remains similar to its template HCE1, with the same fold and a small root mean square deviation between the Cα atoms, and also remains similar in the course of MD simulations (see Appendix A), without any domain unfolding or other significant conformational changes rendering this model as stable. However, our MD simulations indicate an unstable orientation of the residues of the catalytic triad (Ser202, Glu319, and His431), which is also observed in the MD simulations of the HCE1 apo protein. The latter model was built based on an experimental crystal structure. We are therefore confident that these changes in the catalytic triad are not an effect of a poor homology model but intrinsic to the catalytic triad. The “distorted” catalytic triad seems to disagree with the crystal structure of HCE1 that shows Ser221, His467, and Glu353 all in hydrogen bond distance, i.e., perfectly aligned for activation of the serine nucleophile by proton transfer. The MD simulations, however, show a slightly different situation. That is, the protein is in solution at ambient temperature, which could explain the larger flexibility and hence less stable catalytic triad. In fact, the solution structure of another serine protease, PB92 [79], exhibits a “broken” catalytic triad (Ser215, His62, and Asp32), in which the aspartate residue in particular is rotated away from the other residues of the triad in some of the conformations. The crystal structure of the same protein [80], in contrast, exhibits a conformation of the three residues that is compliant with hydrogen bonds between them, similar to the X-ray structure of HCE1. It is therefore conceivable that the “broken” catalytic triad observed in our MD simulations of HCE1 and HCE2 is indeed representative of the proteins in a solvent. For both proteins HCE2 and HCE1, the catalytic triad is significantly more stable when Ser202 is deprotonated and His is protonated (SEM–HIP), that is, after the initial proton transfer has occurred, but still different from the crystal structure of HCE1. The NMR experiments on which the solution structure of the alkaline protease PB92 is based [79] suggest a protonated serine and neutral histidine in its apo state. Therefore, the (SER–HID) protonation state of HCE1 and HCE2 seems to be more likely, at least prior to ligand binding, despite the higher probability of an intact catalytic triad in the (SEM–HIP) models.

A related crystal structure of the HCE1 protein bound to a ligand (PDB-ID 1YA4) shows no significant difference in the active site to the crystal structure of the apo protein, suggesting that large conformational changes or a rearrangement of the catalytic residues upon ligand binding are unlikely in that protein [11,21]. In our MD simulations of different ligands bound to the HCE2 protein, the overall architecture of the binding pocket remains indeed similar to that in the apo protein. The residues exhibit fluctuations that are generally comparable in the apo protein and in the ligand-bound proteins in both protonation states (neutral Ser202 and neutral His431 (SER–HID) and deprotonated Ser202 and protonated His431 (SEM–HIP)). Only His431 fluctuates more, but also with larger error, in the (SER–HID) model bound to irinotecan.

The MD simulations of the HCE2–irinotecan complexes, (SER–HID) and (SEM–HIP), exhibit distances between the residues of the catalytic triad that are similar to those in the respective apo protein, which are, in both cases, significantly shorter when Ser202 is deprotonated and His431 is protonated (SEM–HIP). In contrast, when the ligands L2, L5, and L8 are bound, these distances do not change significantly between apo and ligand-bound complexes in either protonation state of the HCE2 protein. It is likely that the apo protein has a neutral Ser202 and His431, but after ligand binding, both protonation states are conceivable.

The binding poses of the ligands do not vary much between SER–HID and SEM–HIP models. In all cases, the amino group of the ligands is close to a cluster of glutamate residues in the innermost part of the binding pocket, and the camptothecin(-like) moiety is oriented towards the exterior. All ligands are well anchored by the glutamate cluster owing to strong interactions of the protonated amino group with the glutamate residues (Glu201, Glu227, and Glu434) buried in the binding pocket of the protein. Also, the tertiary amine in the irinotecan ligand exhibits favourable interactions with Glu201 in particular. Deprotonated and hence negatively charged Ser202 exhibits favourable electrostatic interactions with the ligands which are, however, roughly compensated by the unfavourable electrostatic interactions between the ligands and protonated His431. From those interactions, it is not clear which of the two protonation forms favours ligand binding (irinotecan or L2, L5, L8) in a reaction-competent position.

With regard to a possible attack by Ser202 as a nucleophile, the position of the ester group is not fully optimal, as quantified by the average distance to Ser202 up to ∼10Å for any of the ligands bound when Ser202 is neutral. This also applies to the substrate ligand irinotecan. It is interesting to note that distances of ∼10Å between the nucleophilic serine and the ester group of irinotecan are also observed in the crystal structure of irinotecan bound to achetylcholine esterase (PDB-ID 1U65 [81]), of which it is an inhibitor. These large distances to Ser202 thus render nucleophilic attack, and hence hydrolysis, infeasible. In contrast, irinotecan bound to HCE2 with deprotonated Ser202 and protonated His431 (SEM–HIP) is not only positioned at a distance but also in an orientation with respect to Ser202 that would allow its attack as a nucleophile and hence initiate ester hydrolysis. Since irinotecan is a known substrate of HCE2 [2,6,12,20], such a pose has to be considered reaction-competent. The other ligands, L2, L5, and L8, exhibit distributions of distances between Ser202 and the ester C-atom that are rather similar for the three ligands, despite their different linker lengths. This is realised through interactions with different residues of the glutamate cluster closer to or farther from the catalytic triad, allowing accommodation of all L2, L5, and L8 ligands in the binding pocket. The variability in the glutamate residue to interact with the amino group of the L2, L5, and L8 ligands, however, results in different placements of the ester group and hence in varying distances to Ser202, even for the same ligand. All ligands exhibit a non-negligible probability for Ser202–ester distances at about ∼6Å, which may result in some, though rather limited, activity of HCE2 on those pro-drug ligands (L2, L5, and L8). In light of pro-drug design, such limited ester hydrolysis by the HCE2 enzyme is desirable.

The initial docking poses of the pro-drug ligands L2, L5, and L8 subjected to MD simulations all exhibit distances between the nucleophilic Ser202 and the ester carbon atom that suggest a reactive complex (see Appendix A). However, these poses are not maintained in the molecular dynamics simulations, that is, when protein and ligand flexibility, as well as the effects of solvent and a finite temperature, are taken into account. For the irinotecan control, in contrast, a pose with the ester carbon atom close to Ser202 is maintained, as anticipated for a substrate of the HCE2 enzyme. Our work thus shows that it is important to go beyond a static approximation of modelling complexes by docking ligands to a crystal structure (if available) and test the stability of such a complex structure by further simulations, such as MD simulations, which allows for a statistical analysis of important interactions.

Revealing such interactions by modelling and simulating enzmye–pro-drug complexes can also help design and improve pro-drug ligands. For example, a variant of the presented ligands L2, L5, and L8 that does not contain a primary amino group at one end may not even bind well to the HCE2 enzyme due to a lack of favourable interactions with the glutamate cluster. Bulkier substituents may have a similar effect. In contrast, a similar ligand with a much shorter distance between the protonated amino group and the ester group, e.g., without a triazole group, would allow binding in a way that has the ester carbon atom poised for attack by the nucleophilic Ser202 and would therefore be processed by HCE2, which is to be avoided. A detailed study of how a pro-drug binds to an abundant enzyme that may catalyse the reaction, releasing the actual drug, is thus valuable to estimate the possible side effects and, with that, the specificity of drug release for its target environment.

## 5. Conclusions

Our homology model of HCE2 remains in a stable fold throughout our MD simulations and retains its similarity with the template protein HCE1. From a comparison of the MD-simulated HCE2 protein in its apo form with the template protein HCE1 in its crystal structure and simulated in solution, we find that the catalytic triad in solution exhibits a larger flexibility than observed in the crystal. This behaviour is also found for models that mimic the protein after the initial proton transfer from Ser202 to His431, and persists upon ligand binding.

Our modelling and molecular dynamics simulations of pro-drug ligands which are activated as drugs by ester hydrolysis reveal similar binding modes to esterase HCE2 to those of the substrate ligand irinotecan. Interactions with a glutamate cluster in the interior of the protein binding pocket are strongly favoured by the protonated amino group of the ligands. The distance of the putative nucleophile Ser202 to the ester group of all the pro-drug ligands varies between poised and too far away for attack by Ser202. This is in contrast to the distance between Ser202 and the ester group of the substrate irinotecan, which is short enough to render attack by Ser202, and thus hydrolysis, feasible, once Ser202 is deprotonated. For the pro-drug ligands, no significant effect of a shorter distance to the activated, deprotonated Ser202 was observed. The pro-drug ligands bind to the esterase protein; it is therefore unlikely that they are hydrolysed by the enzyme with a considerable activity.

Our study shows, at least for this HCE2 enzyme–ligand system, that a static picture provided by X-ray data and used in most docking studies is not sufficient, and that it is advisable to augment the model by using extensive MD-based techniques. Together, these data can provide insights into the stability of a pro-drug molecule with respect to unsolicited conversion into the active drug by abundant enzymes. The ligands L2, L5, and L8 studied in this work are, according to our data, promising pro-drug candidates, since they are unlikely to be hydrolysed by HCE2. Actual drug release can therefore take place preferably, as desired, in the target cancer cell environment.

## Figures and Tables

**Figure 1 biomolecules-14-00153-f001:**
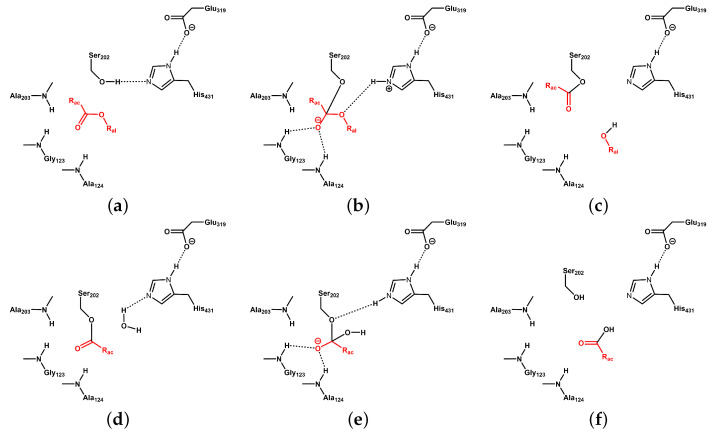
Schematic reaction mechanism of ester hydrolysis by HCE2, according to [3,4]. The substrate and products are highlighted in red; protein residues are shown in black. (**a**) Carboxyl ester substrate with acyl (R_ac_) and alcohol (R_al_) moieties, poised for hydrolysis in the active site of HCE2. (**b**) Tetrahedral intermediate formed by attack of deprotonated Ser202 on the carbonyl atom after proton transfer from Ser202 to His431. The oxyanion is stabilised by hydrogen bonds to Gly123 and Ala124 and further interactions with Ala203. (**c**) Ser202-acyl intermediate and eliminated alcohol (R_al_-OH), protonated by His431. (**d**) Attack of a water molecule, activated by proton transfer to His431, at the carbonyl atom of the acyl intermediate. (**e**) Second tetrahedral intermediate. (**f**) Eliminated carboxyl acid (R_ac_-COOH) and reprotonated Ser202.

**Figure 2 biomolecules-14-00153-f002:**
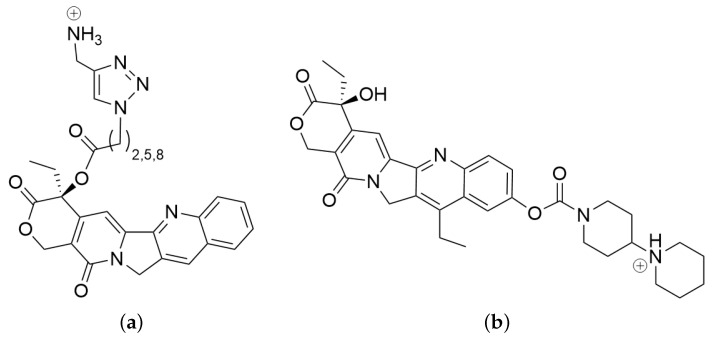
Ligands: (**a**) ester derivatives of camptothecin as putative pro-drugs Ln = L2, L5, and L8, and (**b**) irinotecan (It). Camptothecin and SN38 (7-ethyl-10-hydroxycamptothecin) are the relatively large alcohol moieties preferred by a carboxylesterase of type 2 and the remainder is regarded as a (relatively) small acyl group. Note that the ester group of irinotecan is located at the other end of the camptothecin-like ring system than in amino-triazol-alkyl-acid campthothecin esters L2, L5, and L8.

**Figure 3 biomolecules-14-00153-f003:**
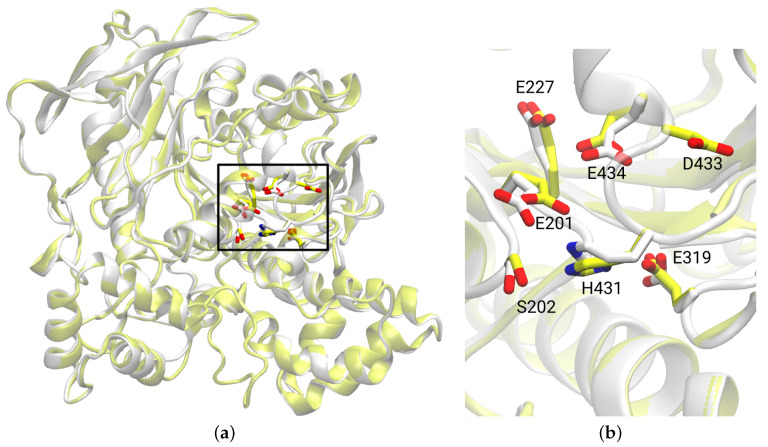
(**a**) Overlay of human carboxylesterase HCE1 (yellow, from PDB structure 5A7H [11]) and homology model of HCE2 (white) with a sequence identity of 48.3% and a root mean square deviation of 0.42 Å. Residues of the active site are highlighted and numbered according to HCE2. (**b**) Zoom into the active site as indicated by the black rectangle in (**a**). The side chains of residues Ser202, His431, and Glu319 of the catalytic triad and residues Glu227, Asp433, and Glu434 are highlighted in stick representation.

**Figure 4 biomolecules-14-00153-f004:**
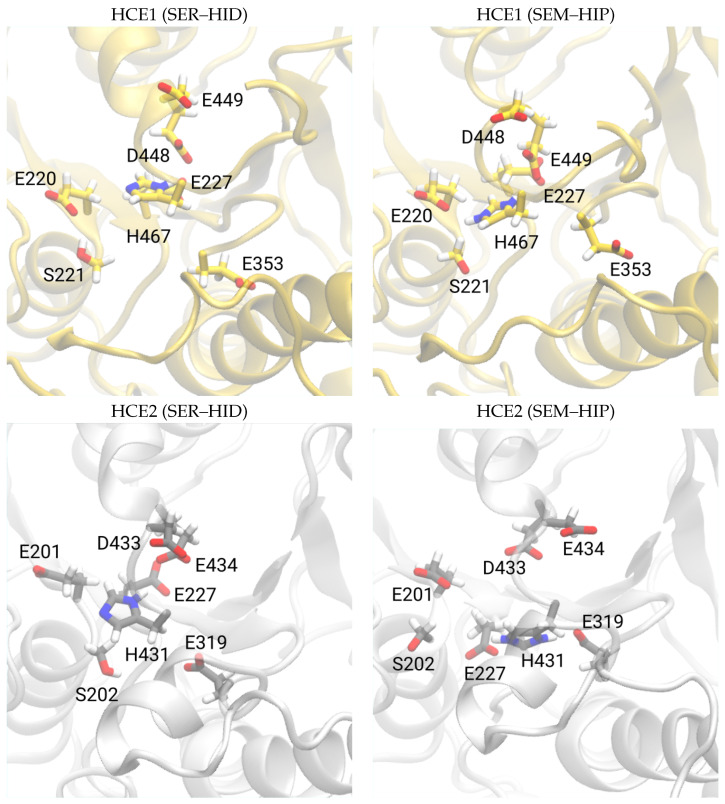
Catalytic triad in the apo esterase proteins HCE1 and HCE2 for comparison. The side chains of active site residues are highlighted as sticks and labelled. SEM–HIP refers to proteins with deprotonated serine (Ser200 and Ser202 in HCE1 and HCE2, respectively) and protonated histidine (His467 and His431 in HCE1 and HCE2, respectively) in the catalytic triad. SER–HID refers to residues modelled as neutral. For the snapshots from all five individual simulation runs per model, see Appendix A.

**Figure 5 biomolecules-14-00153-f005:**
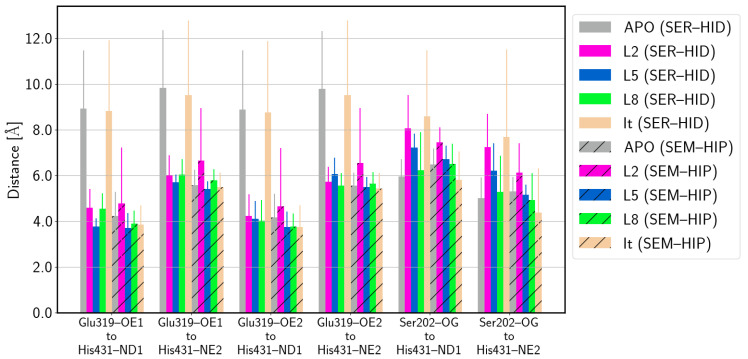
Average distance between the residues of the catalytic triad in HCE2 in apo form and complexed to the ligands. (SER–HID) and (SEM–HIP) models are shown without and with hatching, respectively. For the time series of distances in the apo proteins and the complexes, see Appendix A.

**Figure 6 biomolecules-14-00153-f006:**
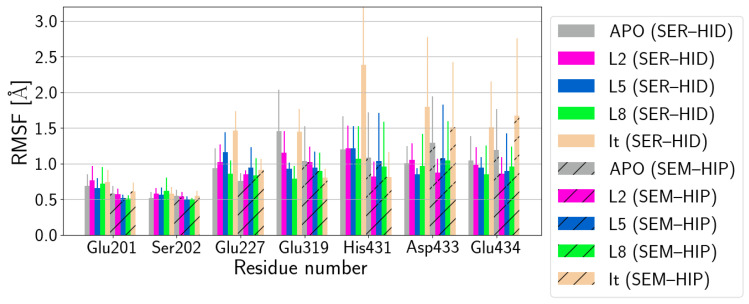
Root mean square fluctuations (RMSF) of selected protein residues: Glu201, Ser202, Glu227, Glu319, His431, and Glu434. (SER–HID) and (SEM–HIP) models are shown without and with hatching, respectively. For comparison of the fluctuations of HCE1 and HCE2 in apo form, see Appendix A. For the fluctuations of the entire protein, see Appendix A. Fluctuations of the ligands are reported in Appendix A.

**Figure 7 biomolecules-14-00153-f007:**
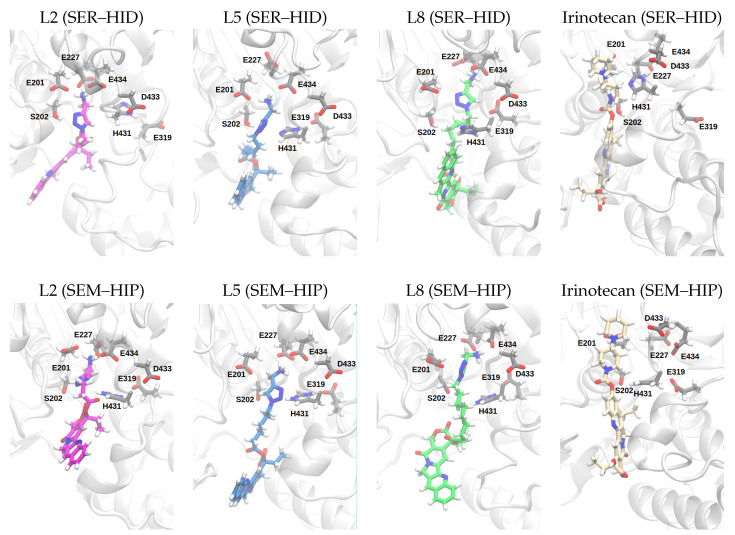
Snapshots (medoid structures, i.e., the structure that is closest to the respective mean structure) from one of the MD simulations of the ligands bound to HCE2 (for snapshots of the other MD simulation runs, see Appendix A). (SEM–HIP) refers to the esterase with deprotonated Ser202 and protonated His431; (SER–HID) refers to neutral Ser202 and His431, respectively.

**Figure 8 biomolecules-14-00153-f008:**
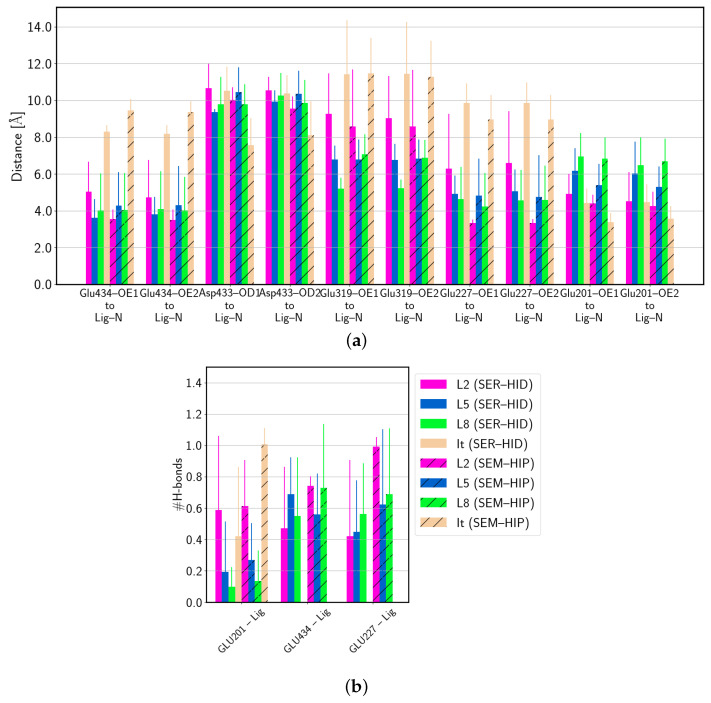
(**a**) Average distance and (**b**) average number of hydrogen bonds between protein residues and the ligands. (SER–HID) and (SEM–HIP) models are shown without and with hatching, respectively. For time series of distances between active site residues of the HCE2 protein and the ligands, see Appendix A.

**Figure 9 biomolecules-14-00153-f009:**
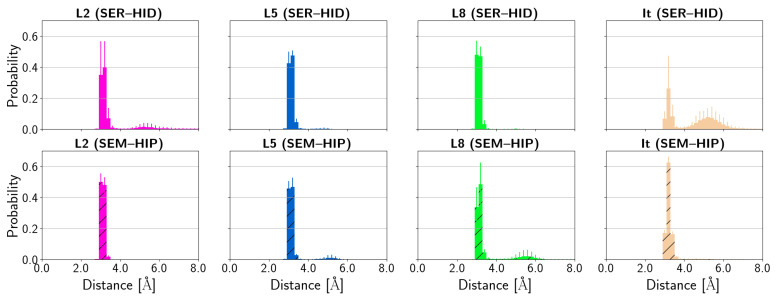
Average minimal distance between glutamate and aspartate residues of the binding pocket and the pro-drug ligands. (SER–HID) and (SEM–HIP) models are shown without and with hatching, respectively. For time series of distances between active site residues of the HCE2 protein and the ligands, see Appendix A. Average distances between Ser202 and other atoms of the ligands are reported in Appendix A; average distances between residues putatively forming the oxyanion hole are shown in Appendix A.

**Figure 10 biomolecules-14-00153-f010:**
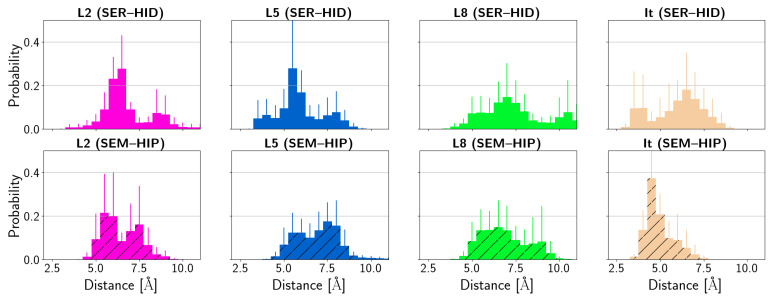
Probability distribution of distances between Ser202 and the ester carbonyl C-atom of the ligands. (SER–HID) and (SEM–HIP) models are shown without and with hatching, respectively.

**Table 1 biomolecules-14-00153-t001:** Average number of hydrogen bonds formed by the side chains of the residues of the catalytic triad (highlighted in bold, numbering according to HCE2). Note that only hydrogen bonds with an average probability of at least 0.5 in at least one of the models are listed for clarity.

(**SER–HID)**							
**Acceptor**	**Donor**	**HCE1 Apo**	**HCE2 Apo**	**HCE2 L2**	**HCE2 L5**	**HCE2 L8**	**HCE2 It**
Glu201	**Ser202**	–	–	0.64 ± 0.44	0.60 ± 0.54	–	0.67 ± 0.43
**Glu319**	Ser228	–	–	–	0.54 ± 0.35	0.90 ± 0.14	–
**Glu319**	Asn316	0.68 ± 0.39	–	0.52 ± 0.47	0.82 ± 0.20	0.88 ± 0.19	–
**Glu319**	Gly317	0.92 ± 0.19	–	–	–	–	–
**Glu319**	**His431**	–	–	0.54 ± 0.40	0.77 ± 0.25	0.54 ± 0.38	–
Glu201	**His431**	–	–	–	–	–	–
Asp433	**His431**	–	–	–	–	0.58 ± 0.32	–
Glu434	**His431**	0.60 ± 0.35	–	–	–	–	0.58 ± 0.45
**(SEM–HIP)**							
**Acceptor**	**Donor**	**HCE1 Apo**	**HCE2 Apo**	**HCE2 L2**	**HCE2 L5**	**HCE2 L8**	**HCE2 It**
Glu201	**Ser202**	–	–	–	–	–	–
**Glu319**	Ser228	–	–	–	–	0.71 ± 0.44	0.66 ± 0.42
**Glu319**	Asn316	–	0.56 ± 0.40	–	0.87 ± 0.06	0.74 ± 0.42	0.68 ± 0.39
**Glu319**	Gly317	0.72 ± 0.45	–	–	–	–	–
**Glu319**	**His431**	1.41 ± 0.94	0.64 ± 0.59	0.76 ± 0.42	0.76 ± 0.32	0.81 ± 0.30	0.66 ± 0.44
Glu201	**His431**	0.66 ± 0.39	–	–	–	–	–
Asp433	**His431**	–	–	–	–	0.58 ± 0.20	–
Glu434	**His431**	–	–	–	–	–	–

**Table 2 biomolecules-14-00153-t002:** Electrostatic interaction energies (kcal/mol) between residues of the binding pocket and the different ligands bound to HCE2. Van der Waals interaction energies are reported in Appendix A.

(**SER–HID)**							
	**Glu201**	**Ser202**	**Glu227**	**Glu319**	**His431**	**Asp433**	**Glu434**
L2	−69.13 ± 18.22	5.95 ± 2.99	−61.17 ± 25.59	−35.31 ± 7.51	−2.50 ± 4.47	−27.97 ± 2.75	−66.86 ± 20.19
L5	−54.59 ± 17.01	5.87 ± 0.88	−72.99 ± 15.48	−51.00 ± 7.85	−1.65 ± 3.04	−31.43 ± 1.75	−85.65 ± 10.24
L8	−47.70 ± 8.39	1.89 ± 4.26	−77.97 ± 18.97	−68.68 ± 13.66	−0.26 ± 3.40	−33.33 ± 3.89	−83.74 ± 22.61
It	−75.92 ± 12.66	2.50 ± 2.32	−37.88 ± 4.77	−34.47 ± 9.04	−5.81 ± 3.67	−34.18 ± 4.03	−41.02 ± 1.63
**(SEM–HIP)**							
	**Glu201**	**Ser202**	**Glu227**	**Glu319**	**His431**	**Asp433**	**Glu434**
L2	−76.43 ± 6.08	−41.47 ± 7.73	−94.38 ± 3.10	−34.14 ± 9.58	39.19 ± 6.50	−29.85 ± 2.26	−83.74 ± 4.91
L5	−66.40 ± 12.17	−50.82 ± 11.61	−78.65 ± 23.27	−48.48 ± 14.59	48.46 ± 5.94	−28.87 ± 1.74	−78.29 ± 16.88
L8	−51.22 ± 13.39	−42.50 ± 13.49	−81.69 ± 23.10	−43.23 ± 6.39	44.84 ± 7.17	−30.82 ± 2.66	−81.81 ± 21.37
It	−91.58 ± 6.93	−50.67 ± 3.95	−39.69 ± 7.51	−32.88 ± 7.99	41.18 ± 4.97	−48.43 ± 10.31	−37.01 ± 2.11

## Data Availability

Force field parameters of the SEM residue and of the ligands as well as PDB structures of the homology model and the docked complexes are available in the Appendix A.

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
