# Peer review of "In Silico Study of Camptothecin-Based Pro-Drugs Binding to Human Carboxylesterase 2"

_biomolecules, 2024, doi:10.3390/biom14020153_

Round 1

Reviewer 1 Report

Comments and Suggestions for Authors

The authors have done a commendable job in investigating the stability and similarity of the homology model of HCE2. The comparison between the crystal structure of HCE1 and the MD simulated HCE2 protein is well-executed and highlights the flexibility of the catalytic triad in solution. This finding adds an important layer of understanding to the enzymatic activity of HCE2.

The analysis of the distances between the nucleophile Ser202 and the ester group of the ligands is well-described. It is interesting to note that, despite varying distances, the hydrolysis activity of the pro-drug ligands remains relatively unaffected. This finding suggests the need for further investigation into the specific mechanisms of hydrolysis.

It would be beneficial if the authors could provide additional discussion on the broader implications of their findings. How might this research contribute to the design and development of more effective pro-drugs?

In order to thoroughly investigate protein-ligand interactions involving chemical reactions, it is crucial to go beyond traditional molecular dynamics (MD) simulations alone. Therefore, I highly recommend the author to enhance their research by incorporating quantum mechanics/molecular mechanics (QM/MM) simulations. This will provide a more comprehensive analysis and allow for a double-checking of the conclusions.

Comments on the Quality of English Language

The english writing can be improved.

Reviewer 2 Report

Comments and Suggestions for Authors

The manuscript is well written. I do have one question.

Have authors tried Alphafold to look for a predicted model for HCE2?

There are a few typos in the manuscript, please proof read it thoroughly

Comments on the Quality of English Language

Minor English edits required

Round 2

Reviewer 1 Report

Comments and Suggestions for Authors

This work presents a focused study on the potential specificity of pro-drug activation by human carboxylesterase 2 (HCE2) through computational methods, providing valuable insights for the development of cancer-targeted chemotherapy. The use of molecular docking and dynamics simulations to investigate the interaction between HCE2 and pro-drug esters of camptothecin offers a detailed assessment of the activation mechanism. The comparison with irinotecan provides a reference for understanding enzyme-substrate compatibility, as well as the stability of the reactant-competent pose after the key proton transfer, which is crucial for hydrolysis. The work outlined in the abstract underscores the importance of considering the dynamic nature of enzyme-ligand complexes and the impact of environmental factors on pro-drug activation. The conclusion regarding the low likelihood of camptothecin ester hydrolysis by HCE2 in normal tissue suggests a promising pathway toward reducing unintended cytotoxicity, enhancing the appeal of this approach in the continuous effort to refine cancer treatments.

Author Response

We thank the reviewer for the very positive feedback to our work.